# Specific Signal Transduction of Constitutively Activating (D576G) and Inactivating (R476H) Mutants of Agonist-Stimulated Luteinizing Hormone Receptor in Eel

**DOI:** 10.3390/ijms24119133

**Published:** 2023-05-23

**Authors:** Seung-Hee Choi, Munkhzaya Byambaragchaa, Dae-Jung Kim, Jong-Hyuk Lee, Myung-Hwa Kang, Kwan-Sik Min

**Affiliations:** 1Animal BioScience, School of Animal Life Convergence, Hankyong National University, Ansung 17579, Republic of Korea; 2Institute of Genetic Engineering, Hankyong National University, Ansung 17579, Republic of Korea; 3Aquaculture Industry Division, South Sea Fisheries Research Institute, National Institute of Fisheries Science (NIFS), Yeosu 59780, Republic of Korea; 4College of Pharmacy, Chung-Ang University, Seoul 06974, Republic of Korea; 5Department of Food Science and Nutrition, Hoseo University, Asan 31499, Republic of Korea; 6Carbon-Neutral Resources Research Center, Hankyong National University, Ansung 17579, Republic of Korea

**Keywords:** eel LHR, constitutively activating mutation, inactivating mutation, cAMP response, cell-surface loss of receptor

## Abstract

We investigated the mechanism of signal transduction using inactivating (R476H) and activating (D576G) mutants of luteinizing hormone receptor (LHR) of eel at the conserved regions of intracellular loops II and III, respectively, naturally occurring in mammalian LHR. The expression of D576G and R476H mutants was approximately 58% and 59%, respectively, on the cell surface compared to those of eel LHR-wild type (wt). In eel LHR-wt, cAMP production increased upon agonist stimulation. Cells expressing eel LHR-D576G, a highly conserved aspartic acid residue, exhibited a 5.8-fold increase in basal cAMP response; however, the maximal cAMP response by high-agonist stimulation was approximately 0.62-fold. Mutation of a highly conserved arginine residue in the second intracellular loop of eel LHR (LHR-R476H) completely impaired the cAMP response. The rate of loss in cell-surface expression of eel LHR-wt and D576G mutant was similar to the agonist recombinant (rec)-eel LH after 30 min. However, the mutants presented rates of loss higher than eel LHR-wt did upon rec-eCG treatment. Therefore, the activating mutant constitutively induced cAMP signaling. The inactivating mutation resulted in the loss of LHR expression on the cell surface and no cAMP signaling. These data provide valuable information regarding the structure–function relationship of LHR–LH complexes.

## 1. Introduction

Lutropin/choriogonadotropin receptors (LHRs) belong to the family of G protein-coupled receptors (GPCRs), the largest group of membrane proteins. Luteinizing hormone (LH) is a glycoprotein hormone secreted from the pituitary gland and is an essential regulatory element of reproduction in mammals and fish. Its receptor, LHR, together with the follicle-stimulating hormone receptor (FSHR) and thyroid-stimulating hormone receptor (TSHR), constitute the glycoprotein hormone receptor group and belong to the largest group of membrane proteins [1,2]. *LHR* is associated with numerous mutations related to reproductive failure in humans and mice [3,4,5,6,7,8].

Activating mutations cause familial male-limited gonadotropin-independent precocious puberty (FMPP) and testotoxicosis [9]. Naturally occurring mutations associated with *LHR* cause reproductive disorders in mammals [10]. Depending on the impact on receptor signaling, activating and inactivating mutations can occur independently of hormonal effects. Activating mutations in *LHR* have been reported in males with sporadic or common FMPP [7,11,12]. Inactivating mutations in *LHR* may cause Leydig cell hypoplasia, a rare form of 46XY disorder in sex development [13,14]. They are also reported as a novel cause of hereditary amenorrhea [9] and infertility in females [15,16]. The naturally occurring human (hLHR)-D564G (equivalent to D576G in eel LHR) mutant displays constitutive activation, thereby leading to an increase in basal cAMP response without agonist treatment [17]. The rat (rLHR)-D556G (equivalent to D576G in eel LHR) mutant with a constitutively activating site has been reported, demonstrating the trafficking of 92-kDa mature form of the receptor to the trans-Golgi [18]. The cell-surface loss of the receptor for an activating equine mutant LH/CGR (eLH/CGR)-D564G (equivalent to D576G in eel LHR) was faster than that of wild-type (wt) eLH/CGR, indicating a significant increase in the basal cAMP response [19].

In contrast, the inactivating mutant of rLHR-R442H (equivalent to R476H in eel LHR) has been reported to impair signal transduction upon treatment with a high dose of agonist [20]. The inactivating eLH/CGR R464H (equivalent to R476H in eel LHR) mutant has also been reported to completely impair cAMP response and cause the loss of cell-surface receptors [21]. Inactivating mutations of GPCRs, LHR, and FSHR have been clarified by reviewing the structure–function insights and therapeutic implications [6].

Although several studies have been reported on signal transduction of naturally occurring hLHR, rLHR, and eLH/CGR, there are few reports on the characterization and function of signal transduction of eel LHR in activating/inactivating mutants. We designed the present study to investigate the possibility that the highly conserved activating/inactivating mutants in eel LHR are implicated in signal transduction and loss of cell-surface receptors in cells expressing these receptors.

In the present study, we generated activating (D576G) and inactivating (R476) mutants of eel LHR to elucidate their effects on receptor signaling mechanisms, including receptor activation and cell-surface loss of receptors. In addition, since recombinant equine chorionic gonadotropin (eCG) shows both LH- and FSH-like activities in non-equid species [21], it has potent activity in eel LHR expressing cells, we included rec-eCG as an additional agonist to investigate its effects on signal transduction of eel *LHR* mutants.

## 2. Results

### 2.1. Preparation of Eel LHR Mutants and Their Cell-Surface Expression

We generated one constitutively activating mutant of eel LHR in intracellular loop 3 (D576G) and one inactivating mutation in intracellular loop 2 (R476H) to determine their effects on hormone–receptor interaction (Figure 1). Previous studies on activating and inactivating eel LHR mutants have been conducted in the transmembrane domains II, III, V, and VI. In the present study, these two mutants were located in intracellular loops 2 and 3. The mutated sites were conserved well within LHRs of mammalian and fish species. Next, we analyzed the cell-surface expression of eel LHR mutants in the human embryonic kidney (HEK) 293 cells.

Receptor expression was equivalent in cells expressing the activating mutant or eel LHR-wt. We considered the expression of eel LHR-wt as 100%. The expression of D576G and R476H mutants was approximately 58% and 59%, respectively (Figure 2 left). Although both mutants showed low expression, they were typically expressed on the cell surface. Previously, eLH/CGR R464H was not expressed on the cell surface [19]. We also determined the number of receptor proteins expressed by western blotting. Two major bands were detected, demonstrating that the lower ~70 kDa indicated intracellular eel LHR precursor. The other higher band suggests conversion of the precursor to the cell surface ~90 kDa eel LHR. The bands at ~180 kDa represent receptor dimers seeing around 90 kDa of monomeric receptor forms.

Next, we determined cAMP response and cell-surface loss of receptors induced by agonist treatment.

### 2.2. Expression and Western Blot Analysis of Rec-Eel LH

The plasmid encoding single-chain eel LHβ/α was constructed and transfected into CHO-K1 cells. Rec-eel LH was expressed and quantified as previously described [22]. The molecular weight of rec-eel LH was analyzed by western blotting using a monoclonal antibody against the α-subunit monoclonal antibody raised in our lab [23]. The result revealed an approximate molecular weight of 32 kDa (Figure 3), which was consistent with previous reports on glycosylated rec-eel LH (Figure 3). To confirm the molecular weight by N-linked deglycosylated enzyme treatment, we performed enzymatic digestion of oligosaccharides using PNGase F. The molecular weight of rec-eel LH decreased to 25 kDa, demonstrating that glycosylation accounted for approximately 7 kDa (Figure 3). These results indicated that oligosaccharides were highly modified in glycoproteins produced from CHO-suspension cells.

### 2.3. cAMP Responsiveness Induced by Agonists in Activating and Inactivating Mutants

The effects of activating and inactivating mutations on the basal and rec-eel LH- stimulated cAMP responses are summarized in Figure 4 and Table 1.

The cAMP production in cells transfected with eel LHR-wt plasmid DNA increased in a dose-dependent manner. The basal and Rmax cAMP response values were 2.6 and 65.6 nM/10^4^ cells, respectively. The half maximal effective concentration (EC_50_) of eel LH in stimulating cAMP response was approximately 176.6 ng/mL. However, the basal cAMP responsiveness in cells expressing LHR-D576G increased to 15.1 nM/10^4^ cells without agonist treatment (Figure 5). In contrast to the CHO-K1 cells harboring the wt receptor, cells expressing LHR-D576G exhibited a 5.8-fold increase in basal cAMP production, indicating that the receptor was constitutively active without agonist treatment. The basal cAMP level in cells expressing LHR-D576G corresponded to 23% of the maximal response detected in cells expressing eel LHR-wt. The maximal cAMP response in cells expressing LHR-D576G mutant with respect to that in the wt strain was approximately 0.62-fold (Table 1). The maximal cAMP level (15,000 ng/mL) in cells expressing LHR-D576G showed a cAMP response by agonist treatment (100 ng/mL) almost comparable to that observed in cells expressing eel LHR-wt. Therefore, LHR-D576G-expressing cells did not respond to further stimulation with high concentrations of agonists (Figure 4). In contrast, the cells expressing R476H showed completely impaired basal cAMP production even at high agonist concentrations, and EC_50_ and Rmax values could not be measured (Figure 5).

As a result, we observed a specific increase in basal cAMP response by the cells expressing eel LHR-D576G mutant. However, the inactivating mutant (eel LHR-R476H) completely impaired the cAMP response.

### 2.4. Loss in Cell-Surface Receptor by Agonist Treatment

We used enzyme-linked immunosorbent assay (ELISA) to measure the loss of eel LHR expression on the cell surface to further explore the relationship between cAMP level and loss in cell-surface receptor expression. Cells were preincubated with 250 ng/mL rec-eel LH for 60 min. The data for time-dependent loss of eel LHR expression are presented in Figure 6.

The experimental results are summarized in Figure 6 and Table 2. In cells expressing eel LHR-wt treated with rec-eel LH, cell-surface expression decreased to approximately 65% within the first 5 min and remained approximately between 60–68% for 60 min. For the activating mutant, cell-surface expression decreased to around 60% within the first 5 min, with a further decrease to about 50% after 15 min. For the inactivating mutant, cell-surface expression similarly decreased to eel LHR-wt by approximately 62% within the first 5 min.

Additionally, we analyzed the loss of receptors from the cell surface following treatment with the rec-eCG agonist to determine how it affects cells expressing eel LHRs. In eel LHR-wt expressing cells treated with rec-eCG (250 ng/mL), cell-surface expression of the receptor decreased to approximately 67% within the first 5 min, then decreased slowly and remained at approximately 55% for 60 min. Cell-surface expression of the activating mutant decreased slightly more than that of eel LHR-wt to approximately 61% within the first 5 min and decreased further to approximately 42% at 15 min. In the inactivating mutant, loss of receptor decreased more rapidly than eel LHR-wt by approximately 51% within the first 5 min, further reduced to approximately 38% after 15 min, and slightly increased (Figure 7).

With rec-eel LH treatment as an agonist, loss of the expression of the cell-surface receptor at 30 min considerably decreased for the eel LHR-wt (39%) compared to that observed in control cells (considered as 0% loss of surface receptor). For the D576G and R476H mutants, the cell-surface loss of receptors decreased to 40% and 37% at 30 min, respectively, demonstrating no difference among the groups (Figure 8). Specifically, the loss of cell-surface receptors for inactivating mutants reduced to a ratio similar to that found in eel LHR-wt, despite the completely impaired cAMP response. Loss of surface receptors for eel LHR-wt at 30 min was approximately 44% by rec-eCG treatment, indicating that it decreased faster than observed with rec-eel LH agonist (Figure 8). However, the loss of surface receptors in two mutants (eel LHR-D576G and R476H) was decreased to 52% and 61.6%, respectively. Surprisingly, the loss of cell-surface receptors in the inactivating mutant was higher than that in eel LHR-wt.

The rate of formation of the ligand-receptor complexes induced by constitutively activating and inactivating mutants of eel LHR is presented in Table 2. For both the activating and inactivating mutants, the rate was approximately 0.6 min by agonist eel LH treatment. With rec-eCG treatment, the rates of loss of the cell-surface receptor–agonist complex for the mutants were more rapid (1.9–2.1 min) than for eel LHR-wt (Table 2). Specifically, the R476H mutant, upon rec-eel LH and rec-eCG treatment, exhibited the most rapid rates of loss despite the cAMP response being completely impaired. These data clearly showed that the activating mutant reduced the rate of cell-surface loss of the receptor, demonstrating that the rate is consistent with cAMP responsiveness. However, for the inactivating mutant, the rate of cell-surface loss of the receptor was faster than that of eel LHR-wt.

## 3. Discussion

The present study aimed to determine whether the activation/inactivation mutation of eel LHRs was indispensable to the function of the cAMP signaling pathways and if the loss of cell-surface receptors was caused by high-agonist treatment, using eel LHR as a model. Our results showed that the activating mutation D576G might produce constitutively activating signals for cAMP production and cause the loss in expression of cell-surface receptors. The inactivating mutation R476H completely impaired cAMP signal transduction, but the loss of cell-surface receptors commonly occurs following eel LH and eCG ligand treatments. These mutations stimulate basal cAMP responsiveness and/or attenuate the agonist-induced activation of eel LHR.

The N-linked glycosylation sites of rec-eel LH, -eel FSH, and rec-eCG play an essential role in the biological activity of eel LH [22] and eCG [21,24]. The molecular weight of rec-eel LH produced by CHO-S cells was approximately 32 kDa, and deglycosylation by PNGase F markedly decreased the molecular weight. These results are consistent with our previous studies that indicated modification in molecular weight by approximately 7 kDa while expressed in CHO-K1 cells [25,26]. In this study, the increase in molecular weight of rec-eel LH in CHO-S cells indicated that oligosaccharides in the N-linked glycosylation sites (α-Asn^56^, Asn^79^, and β-Asn^10^) were post-translationally modified.

As predicted from the results described above, the two mutations in the present studies were not fully expressed on the cell surface and intracellular eel LHR precursor. Thus, our results are consistent with those previously reported in hLHR [20], rLHR [1], and eLH/CGR [19]. Therefore, we suggest that conformational change in the mutated receptors could explain why the inactivating mutant (R476H) did not produce cAMP responses because of the low expression. However, the activating mutant (D576G) produced high cAMP signal transduction in spite of the low expression. We suggest that activating mutant is a significant model for determining the cellular mechanisms of eel LHR with/without high agonist treatment. Although we did not check in the double mutants, the mutants may be a low expression of cell surface for conformational change.

Our previous observations have suggested that the active conformations of rLHR [1], eel FSHR [27], eLH/CGR [19], and eFSHR [28] were revealed during stimulation of G proteins and loss of the cell-surface receptor by agonist treatment. In the present study, the activating mutant D576G in the third intracellular loop region exhibited a 5.8-fold increase in basal cAMP response, consistent with the observation that hLHR-D564G (equivalent to D576G in eel LHR) displayed a considerable rise in cAMP response without agonist treatment in HEK 293 cells [17]. In a more detailed analysis, changes of various amino acids (G, A, V, N, L, and F) at this site (D564) of human LHR showed constitutive activation in COS-7 cells (three–five-fold increase in basal cAMP), but the glutamate mutant (negative charge) did not show such an increase, thereby indicating that inactive LHR conformation was maintained [29,30]. Recently, we also reported that the point mutation D566G in eFSHR displayed an 8.6-fold increase in the basal cAMP response in HEK-293 cells [28]. The expression of eel FSHR-D540G (equivalent to D576G in eel LHR) in the third intracellular loop region was substantially increased (23.3-fold) without agonist treatment [27]. Our results are consistent with previous studies on the same site in the third intracellular loop region. In the six transmembrane domain regions, hLHR-D578G (equivalent to D590G in eel LHR) mutant exhibited a 4.5-fold increase [10], a 3.5-fold increase in COS-7 cells [31], and a similar increase in HEK 293 cells [5]. However, a substantial increase of approximately 25-fold in the basal cAMP response in HEK 293 cells has been reported [32]. The basal cAMP response differed slightly from that in cells expressing the D576G mutant. The basal cAMP response of double mutant receptors (D564 and D578) was higher than that observed for the single mutant; however, the maximal cAMP response significantly decreased [30].

Therefore, the eel LHR-D576G mutant constitutively activates cAMP signal transduction without agonist treatment. The D576G mutation lies immediately below the six transmembrane domain regions. Both activating mutations (D576G in the third intracellular loop and D590G in the six transmembrane domain regions) are important sites for signal transduction through the receptor–agonist complex. Mutational studies performed with position 576 in eel LHR may reveal conformational changes in the receptor structure, indicating that this residue plays a pivotal role in regulating the conversion of active receptor conformation.

In the activation model, eel LHR-M410T, L469R, and D590Y mutants exhibited a significant increase (4.0–19.1-fold) in basal cAMP response [19]. Eel FSHR-D540G and D540N mutants displayed 14.5–23.2-fold increases without agonist treatment [27]. However, despite prolonged agonist treatment, these activating mutants of eel LHR and eel FSHR did not further increase cAMP accumulation. These results are consistent with our current data, indicating that the eel LHR-D576G mutant constitutively activates the basal cAMP response. Still, the maximal cAMP response is markedly lower than that of eel LHR-wt. Therefore, the reason behind the maximal response of the activating mutants being lower than that of the eel LHR-wt receptors despite the cell-surface expression being approximately 50% relative to the eel LHR-wt needs to be analyzed in detail.

Unlike the activating mutant, the inactivating mutant (eel LHR-R476H) in the second intracellular loop did not elicit any cAMP response despite treatment with a high concentration of ligands and completely impaired signal transduction. These results are consistent with previous reports for the same sites in the second intracellular loop of rLHR-R442H [20] and eLH/CGR-464H [19]. Understanding why the R476H mutant attenuates cAMP response signaling is difficult. The highly conserved arginine residue at position 476 in the second intracellular loop plays a pivotal role in cAMP signal transduction. Our results are consistent with those of other studies on inactivating mutants of eel FSHRs (I193V, N195I, R546C, and A548V) [27] and eFSHR (A189V, N191I, R572C, and A574V) [28], indicating a complete impairment of the cAMP response. Even though we did not analyze it, the cAMP response would be at a completely low level in the double mutants of activating and inactivating eel LHR.

GPCRs are internalized by endosomes via a clathrin-dependent pathway and are then degraded in lysosomes or recycled to the cell membrane for prolonged agonist stimulation [33,34]. In this study, the rates of cell-surface loss of receptor of the activating mutant (0.6–2.1 min) were faster (1.5-fold) than those observed in cells expressing the eel LHR-wt (0.9–2.5 min) by treatment with eel LH and eCG agonists, indicating a definite correlation between basal cAMP response and loss of cell-surface receptor. Specifically, the inactivating mutant did not produce cAMP signaling, but the t_1/2_ value indicated a faster loss of the cell-surface receptor, which are conflicting results. These differences between cAMP signaling and loss of cell-surface receptors need to be clearly demonstrated. These results are consistent with the rLHR-R442H mutant, which displayed a 1.5–2-fold increase in t_1/2_ of internalization [20]. Among the inactivating mutants of eel FSHR, only one (N195I) showed a loss of cell-surface receptor faster (approximately three-fold) than that of the wt eel FSHR [27].

The reason behind the speedy loss of cell-surface receptors in inactivating mutants needs to be elucidated. Interestingly, the loss of cell-surface receptors was faster despite the complete impairment of the cAMP response. Our results do not indicate a possible correlation between the basal cAMP response and the loss of cell-surface receptors. As reported in our previous study, this mechanism is not well-understood. We hypothesized that the inactivating mutant R464H was probably routed to a lysosomal degradation pathway and was not recycled to the cell surface [27]. These results are inconsistent with our previous studies on inactivating mutants of eLH/CGR-R464H [19], eel FSHR-A193V, R546C, A548V [27], and eel FSHR-N191Q [35], where the loss of cell-surface receptors in inactivating mutants did not occur.

In this presented study, we did not determine the internalization in cells expressing both active and inactive receptors. However, we suggest that the receptor loss from the cell surface could help predict the internalization rate into the endosome. One possible mechanism by which a constitutively active mutant of eel LHR might cause increased internalization of the LHR–LH complex has been reported for hFSHR [3]. The cells expressing hLHR-activating mutants showed a more pronounced loss of cell-surface receptors than those expressing hLHR-wt, demonstrating that receptor recycling promotes the maintenance of cell-surface receptors and responds to agonist stimulation [36]. Therefore, the signaling mechanisms of GPCRs are considered to be involved in the cell-surface expression of receptors, loss of cell-surface receptors, internalization, downregulation, and recycling [37,38,39,40]. Thus, we must accurately resolve these unexplained problems by activating or inactivating eel LH receptors. Research is currently underway to prove our theory about the internalization, degradation, and recycling of GPCR. Mutations in LHRs and FSHRs could affect receptor trafficking to the plasma membrane and cause fertility disorders. Thus, the role of LHRs in male gonad development is crucial. A significant challenge from a pharmacologic point of view is how to diminish cAMP signaling by constitutively activating LHR without agonist treatment.

This suggests that agonist-mediated activation of eel LHR is regulated by the expression and loss of receptors on the cell surface for the activating mutant. However, results with the inactivating mutant in the present study showed inconsistencies between the cAMP response and cell-surface loss of receptors. This is otherwise the case for most GPCRs. The signaling mechanisms of eel LHR mutants are yet to be thoroughly investigated. Our results indicate that further studies are necessary to elucidate the signaling mechanisms of glycoprotein hormone receptors in fish.

## 4. Materials and Methods

### 4.1. Materials

Oligonucleotides were synthesized by GenoTek (Daejeon, Republic of Korea). The pGEM-T Easy cloning vector was purchased from Promega (Madison, WI, USA). The mammalian expression vector, pCORON1000 SP VSV-G, was purchased from Amersham Biosciences (Piscataway, NU, USA). The pcDNA3 expression vector, Lipofectamine-2000, FreeStyle MAX transfection reagent, and FreeStyle CHO-S cells were obtained from Invitrogen (Carlsbad, CA, USA). OptiMEM and Ham’s F-12 media were purchased from Gibco BRL (Grand Island, NY, USA). CHO-K1 and HEK 293 cells were obtained from the Korean Cell Line Bank (KCLB, Seoul, Republic of Korea). The cAMP HTRF Assay Kit was purchased from Cisbio (Codolet, France). The monoclonal antibody (5A11) used in the ELISA analysis was produced in our lab, as previously reported [23]. The deglycosylation PNGase kit was purchased from New England Biolabs (Ipswich, MA, USA). The 11A8 and 5A11 monoclonal antibodies were labeled with horseradish peroxidase (HRP) by Medexx, Inc. (Seongnam, Republic of Korea). Fetal bovine serum (FBS) was obtained from HyClone Laboratories (Logan, UT, USA). The QIAprep-Spin plasmid kit was purchased from Qiagen Inc. (Hilden, Germany). Endonucleases and PCR reagents were purchased from Takara Bio, Inc. (Shiga, Japan). Glass spinner and disposable flasks were provided by Corning Inc. (Corning, NY, USA). All other reagents were purchased from Sigma-Aldrich (St. Louis, MO, USA).

### 4.2. Production and ELISA Analysis of Recombinant Eel LH (Rec-Eel LH)

The expression vectors were transfected into CHO-S cells for rec-eel LH production using a previously described method [28]. Briefly, one day before transfection, cells were passaged at 5 × 10^5^ cells/mL. On the day of transfection, plasmid DNA (260 μg) and FreeStyle MAX reagent (260 μL) were diluted with Opti-PRO SFM in a total volume of 8 mL, incubated for 10 min, and then added to the cells (200 mL). Cells were incubated at 37 °C in a humidified atmosphere of 8% CO_2_ on an orbital shaking platform rotating at 135 rpm. Finally, the culture media was collected on day 7 after transfection and centrifuged at 100,000× *g* for 10 min at 4 °C. The supernatants were collected and concentrated using either a Centricon filter or freeze-drying and then mixed with phosphate-buffered saline (PBS). The concentration of rec-eel LH was determined using ELISA, which had been previously standardized in our laboratory [24].

### 4.3. Enzymatic Deglycosylation and Western Blotting of Rec-Eel LH

For western blot analysis, the concentrated sample (40 μg) was mixed with 1 μL of glycoprotein denaturing buffer and boiled at 100 °C for 10 min. After cooling on ice for 10 min, 2 μL GlycoBuffer 2, 2 μL 10% NP-40, 1 μL PNGase F, and distilled water were added to the sample to obtain a total reaction volume of 20 μL. The samples were incubated at 37 °C for 1 h. A 20 μg rec-eel LH sample and deglycosylated rec-eel LH were mixed with 2 × loading buffer, boiled at 100 °C for 10 min, and then cooled on ice for 10 min. The protein samples were separated by 12% sodium dodecyl sulfate-polyacrylamide gel electrophoresis (SDS-PAGE) for 2 h. The proteins were transferred onto a polyvinylidene difluoride membrane using a Bio-Rad Mini Trans-Blot electrophoresis cell (Hercules, CA, USA). The membrane was incubated for 1 h with shaking in a 5% blocking solution in TBS-T (20 mM Tris-HCl, pH 7.6, 140 mM NaCl, 0.1% Tween 20) at room temperature. The membrane was incubated overnight at 4 °C with monoclonal anti-eel α-subunit antibody 11A8 diluted at 1:1500. The membrane was washed three times for 20 min with TBS-T and incubated with goat anti-mouse IgG-HRP secondary antibody (diluted at 1:3000) for 1 h at RT. The membranes were then washed three times, and protein bands were detected using an enhanced chemiluminescence system.

### 4.4. Site-Directed Mutagenesis and Vector Construction

As previously described, mutations were generated using overlap extension polymerase chain reaction (PCR) [27]. Two different sets of PCRs were performed. Full-length PCR products were purified and cloned into a pGEM-T easy vector. After DNA sequencing, the mutant cDNAs were subcloned into the eukaryotic expression vector pCORON1000 SP VSV-G using Xho1 and EcoRI sites. We constructed the following receptor genes: eel LHR-wt, D557G, and R476H. A schematic representation of the mutation sites for activating (D576G) and inactivating (R476H) mutations in eel LHR is shown in Figure 1.

### 4.5. Transient Transfection

CHO cells were cultured in Ham’s F-12 medium containing 50 U/mL penicillin, 50 μg/mL streptomycin, 2 mM glutamine, and 10% FBS. HEK 293 cells were cultured in Dulbecco’s Modified Eagle Medium containing 10 mM HEPES, 50 μg/mL gentamycin, and 10% FBS). One day before transfection, CHO-K1 (2 × 10^5^) and HEK 293 (5 × 10^5^) cells were seeded in 6-well plates. On the day of transfection, DNA was diluted with Opti-MEM combined with Lipofectamine 2000 reagent, and the mixture was incubated for 20 min at room temperature. Cells were washed twice, and the DNA–Lipofectamine complex was added to each well. After 5–6 h, a medium containing 20% FBS was added to each well. CHO cells were used for cAMP analysis 48 h after transfection. HEK 293 cells were used to investigate the loss of surface receptors.

### 4.6. SDS-Polyacrylamide Gel Electrophoresis and Western Blotting of Eel LH Receptors

The transfected cells were solubilized in RIPA buffer (50 mM Tris-HCl, pH 8.0, 150 mM NaCl, 0.1% SDS, 1% Triton X-100, 0.5% sodium deoxycholate) with protease inhibitors. The extracted proteins were separated into 10% polyacrylamide gels. After SDS-PAGE, the proteins were transferred onto a polyvinylidene difluoride membrane for 90 min in a Mini Trans-Blot electrophoresis cell. The membrane was blocked with a 5% blocking buffer. The membrane was reacted with a primary antibody, a rabbit anti-VSV-G tag monoclonal antibody (Cell Signaling; 93372), diluted × 1000 times with a blocking solution overnight at 4C. The membrane was washed three times with TBS-T and incubated with goat anti-rabbit IgG-HRP secondary antibody diluted 1:3000 with the blocking buffer for 1 h at RT and washed by blocking solution. The membrane was then incubated for 5 min with 2 mL of the Lumi-Light substrate solution. The membrane was covered with a second piece of plastic wrap, and an X-ray film was exposed to the membrane for 30 min.

### 4.7. The cAMP Analysis by Homogeneous Time-Resolved Fluorescence (HTRF)

The cAMP accumulation in CHO-K1 cells expressing eel LHR-wt and activating and inactivating mutants was measured using a cAMP Dynamic 2 competitive immunoassay kit (Cisbio Bioassays, Codolet, France), as previously described [19]. Briefly, the assay was conducted using two cAMP antibodies labeled with cryptate-conjugated anti-cAMP monoclonal antibody and d2-labeled cAMP. Transfected cells were diluted in 0.5 mM IBMX to inhibit cAMP degradation and seeded in 384-well plates (10,000 cells per well). Each well was supplemented with 5 μL rec-eel LH, and the plate was incubated for cell stimulation at RT for 30 min. The assay was terminated using detection reagents, cAMP- d2, and anti-cAMP-cryptate (diluted five-fold in lysis buffer, 5 µL/well) for 1 h at RT. The cAMP was detected by measuring the decrease in HTRF energy transfer (665 nm/620 nm) using a Tristar2 SLB942 microplate reader (BERTHOLD Tech, Wildbad, Germany). The specific signal-Delta F (energy transfer) is inversely proportional to the concentration of cAMP in the standard or sample. The results were calculated from the ratio of fluorescence at 665 and 620 nm and expressed as Delta F% (cAMP inhibition) according to the following equation:Delta F% = (standard or sample ratio-mock transfection) × 100/mock transfection.

The cAMP concentrations for Delta F% values were calculated using GraphPad Prism software v.6.0 (GraphPad, Inc., La Jolla, CA, USA).

### 4.8. Agonist-Induced Loss in Cell-Surface Expression of Receptors

The loss of eel LHR at the cell surface was assessed using ELISA, as previously described [19,40]. Cells were plated at a density of 6 × 10^5^ per 60-mm dish and then transfected with eel LHR-wt and mutant constructs. Cells were split into 96-well plates (1 × 10^4^ cells) coated with poly-D-lysine at 24 h post-transfection. The next day, cells were pre-incubated with rec-eel LH (250 ng/mL) and rec-eCG (250 ng/mL) for time-dependent analysis (0, 5, 15, 30, and 60 min). Cells were fixed with 4% paraformaldehyde in Dulbecco’s PBS (DPBS) for 5 min. After washing twice with DPBS, cells were incubated with blocking solution (Tris-buffered saline with 1% BSA) for 30 min, followed by incubation with rabbit anti-VSVG antibody (1:1000) and HRP-conjugated anti-rabbit antibody (1:500) for 1 h. Cells were washed thrice with a blocking solution, and 80 µL DPBS and 10 µL SuperSignal.

ELISA Femto Maximum substrate was added to each well. Luminescence was measured using a Cytation 3 plate reader. The expression level of eel LHR-wt was set to 100% at 0 s. The cell-surface expression levels of wt and mutant eel LHRs were set to 100% in untreated cells. The expression of cell-surface receptors was calculated by comparing loss during agonist stimulation to the levels in untreated cells (taken as 0% of the loss of cell-surface receptors).

### 4.9. Data Analysis

The Multalin multiple sequence alignment tool was used for sequence analysis. GraphPad Prism 6.0 (GraphPad, Inc.) was used for cAMP production analysis and cAMP EC_50_ values. GraFit 5.0 (Erithacus Software Limited, Surrey, UK) was used for stimulation curve analyses. Curves fitted in a single experiment were normalized with respect to the background signal in mock-transfected cells (Figure 4). The data for the mock-transfected cells were subtracted from the results for cAMP levels and cell-surface receptors in the transfected cells. Each curve was plotted using data from three independent experiments. The results are expressed as the mean ± SD of a single representative experiment performed in triplicate. Data were analyzed using one-way analysis of variance (ANOVA) followed by Tukey’s comparison test with GraphPad Prism v.6.0 and indicated in the figure captions. Statistical significance was determined at the following levels: * *p* < 0.05 and ** *p* < 0.01 indicated a significant difference between groups.

## 5. Conclusions

This study showed that a constitutively activating mutation of eel LHR (D576G) resulted in a significant increase in basal cAMP production. The mutant displayed a faster loss of cell-surface receptors than those observed for eel LHR-wt, despite reduced cell-surface expression. For the inactivating mutant (eel LHR-R476H), cAMP signaling was completely impaired by high agonist stimulation; however, the rate of loss of cell-surface receptor was slightly higher than that of the activating mutant D576G and eel LHR-wt. The findings of this study are crucial to our understanding of LHR function and regulation with regard to mutations of highly conserved amino acids in the intracellular loops of mammalian and/or fish glycoprotein hormone receptors. Future studies on the mutations of glycoprotein hormone receptors should attempt to identify the mechanism underlying the structure-function relationships of eel LHR–LH complexes in the cAMP signaling pathway.

## Figures and Tables

**Figure 1 ijms-24-09133-f001:**
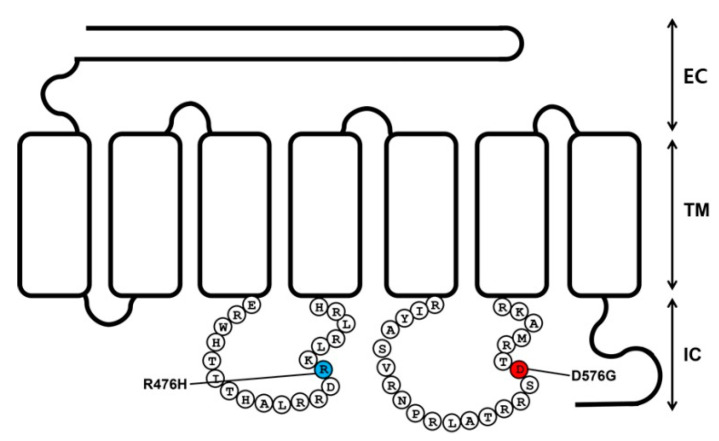
Schematic representation of the structure of eel luteinizing hormone receptor (LHR). The locations of the constitutive activating mutation (D576G) in the third intracellular loop and the inactivating mutation (R476H) in the second intracellular loop are indicated. The red and blue circles indicate the constitutively activating and inactivating mutations. EC, extracellular loop. TM, transmembrane domain. IC, intracellular loop.

**Figure 2 ijms-24-09133-f002:**
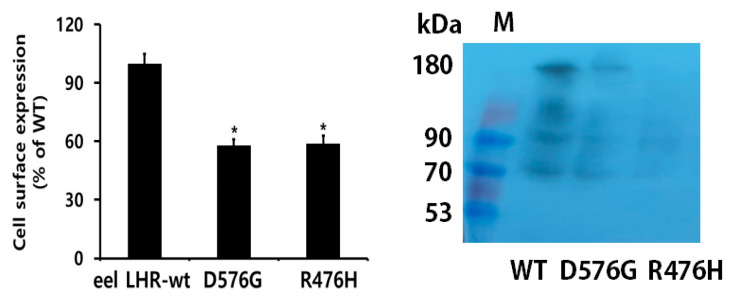
Cell-surface expression of eel LHRs in transiently transfected HEK-293 cells. Enzyme-linked immunosorbent assay (**left**) and western blotting (**right**) were used to determine the surface expression of eel LHRs. Values are means ± SEM for three independent experiments and were normalized with respect to the wild-type eel LHR. Cell-surface expression of the wild-type eel LHR (eel LHR-wt) was considered 100%. * Statistically significant differences from eel LHR-wt were calculated by a one-way ANOVA followed by Tukey’s comparison test (* *p* < 0.05).

**Figure 3 ijms-24-09133-f003:**
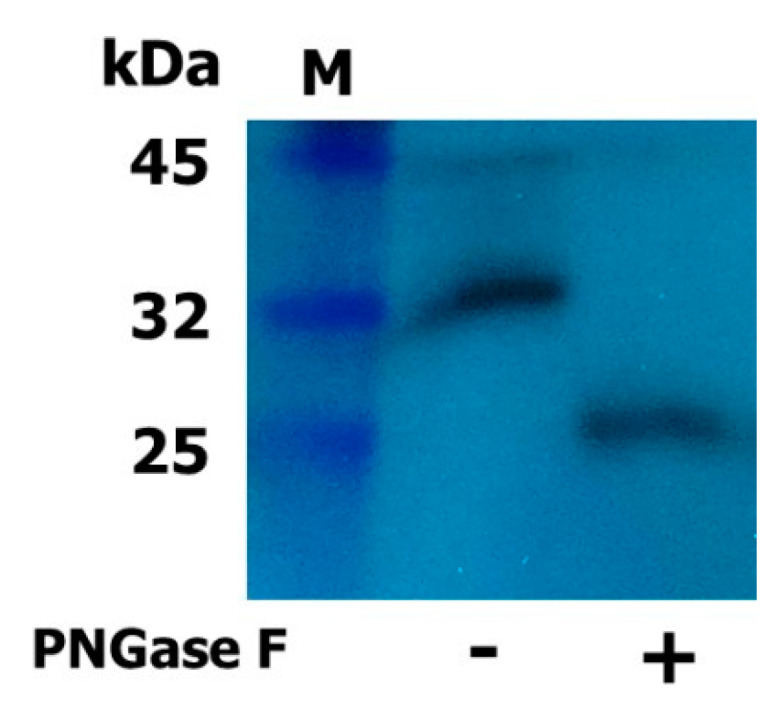
Western blot analysis of recombinant (rec)-eel LH. Rec-eel LH was expressed in CHO suspension cells. Conditioned media were collected, and proteins were separated by sodium dodecyl sulfate-polyacrylamide gel electrophoresis. Proteins were detected using a monoclonal antibody against the eel α-subunit. Protein samples were treated with N-glycosidase-F. PNGase digestion was conducted for 1 h at 37 °C. M: marker; −: untreated; +: treated with N-glycosidase-F.

**Figure 4 ijms-24-09133-f004:**
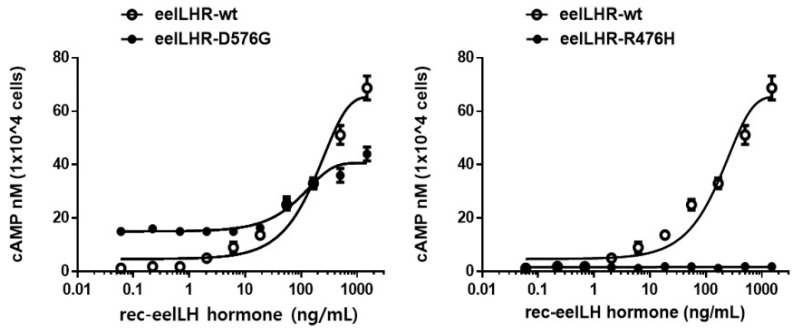
Total cAMP levels stimulated by recombinant eel LH in CHO-K1 cells transfected with constitutively activating (D576G: **left**) and inactivating (R476H: **right**) eel LH receptors. CHO-K1 cells transiently transfected with eel LHR-wt and mutants (D576G and R476H) were stimulated with rec- eel LH in a medium containing 0.5 mM 3-isobutyl-1-methyl xanthine for 30 min before total cAMP was assayed. Levels of cAMP production were determined by homogeneous time-resolved fluorescence. cAMP accumulation was calculated as Delta F%. cAMP concentration was recalculated using GraphPad Prism software v.6.0 (GraphPad, Inc., La Jolla, CA, USA). The values obtained for mock transfection were subtracted from each data set. A representative data set was obtained from three independent experiments. The blank circles represent the same curves of eel LHR-wt.

**Figure 5 ijms-24-09133-f005:**
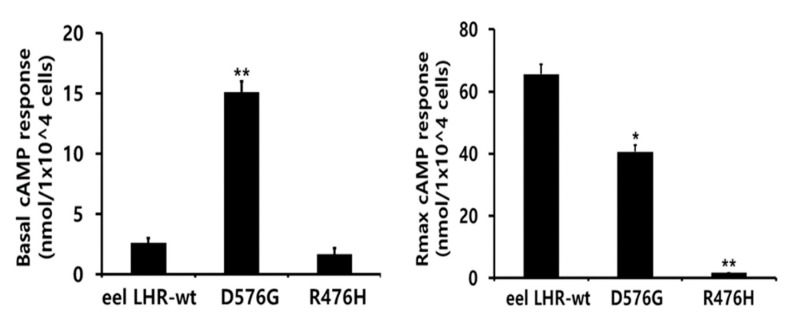
Basal cAMP responsiveness (**left**) and Rmax level (**right**) in activating (D576G:) and inactivating (R476H) mutants. The basal and maximal cAMP responses presented in Figure 4 are displayed using a bar graph. The results are expressed as mean ± SD of a single representative experiment performed in triplicate. * Statistically significant differences from eel LHR-wt in basal cAMP level and Rmax cAMP response were calculated by a one-way ANOVA followed by Tukey’s comparison test (* *p* < 0.05 and ** *p* < 0.01).

**Figure 6 ijms-24-09133-f006:**
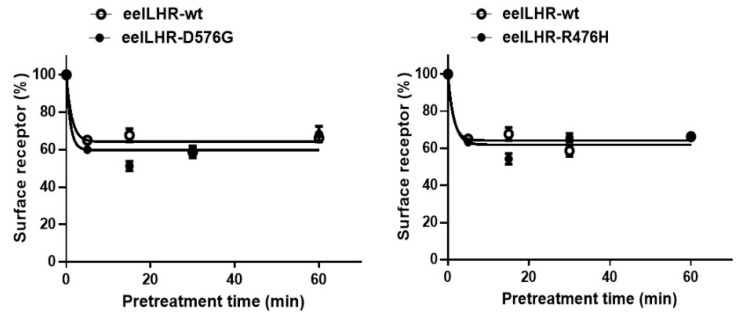
Loss of cell-surface expression of eel LHR-wt and activating (**left**)/inactivating (**right**) mutants. Each plasmid was transiently transfected into HEK-293 cells. Cells were incubated with or without 1000 ng/mL rec-eel LH for 60 min, and the expression of receptors on the cell surface was determined. Results are expressed as a percentage of receptor loss on the cell surface calculated by comparing levels in the presence of rec-eel LH to levels in the absence of agonist (taken as 100% cell-surface expression). The results are expressed as the mean ± SD of a single representative experiment performed in triplicate.

**Figure 7 ijms-24-09133-f007:**
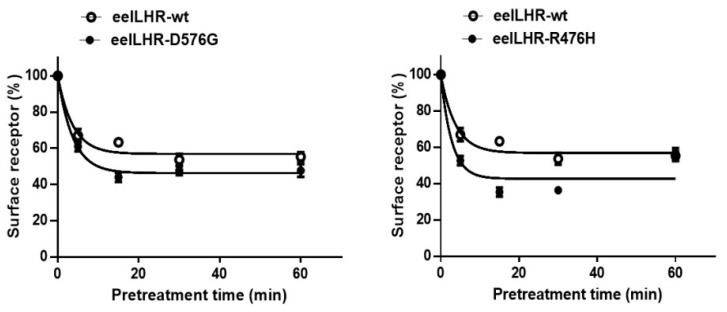
Loss of cell-surface expression of eel LHR-wt and activating (**left**)/inactivating (**right**) mutants. Cells were incubated with or without 1000 ng/mL rec-eCG for 60 min, and LHR expression was calculated by comparing the levels during agonist stimulation to levels in the absence of agonist (considered 100% cell-surface expression). The results are expressed as the mean ± SD of a single representative experiment performed in triplicate. In this figure, mean data were fitted to the one-phase exponential decay equation. The blank circles are the same curves of eel LHR-wt.

**Figure 8 ijms-24-09133-f008:**
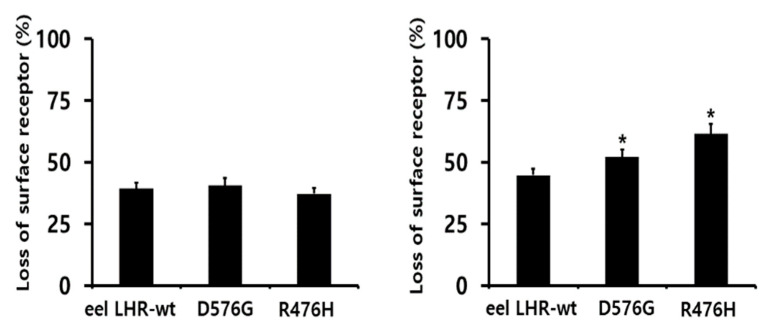
Loss of cell=surface expression of eel LHR-wt and activating/inactivating LHR mutants. HEK-293 cells transiently expressing eel LHR-wt or activating/inactivating receptors were incubated with 1000 ng/mL rec-eel LH (**left**) and rec-eCG (**right**) for up to 60 min. Loss of receptor expression on the cell surface in the non-pretreated groups was considered 0%. The results are expressed as percent loss on the cell surface at 30 min. The loss of each receptor is shown using GraphPad Prism software. The results are expressed as the mean ± SD of a single representative experiment performed in triplicate. * Statistically significant differences from eel LHR-wt were calculated by a one-way ANOVA followed by Tukey’s comparison test (right panel) (* *p* < 0.05).

**Table 1 ijms-24-09133-t001:** Bioactivity of eel LH receptors in cells expressing activating and inactivating receptor mutants.

Eel LH Receptors	cAMP Responses		
	Basal *^a^* (nM/10^4^ Cells)	EC_50_ (ng/mL)	Rmax *^b^* (nM/10^4^ Cells)
LHR-WT	2.6 ± 0.4 (1-fold)	176.6 (138.7 to 242.7) *^c^*	65.6 ± 3.2 (1-fold)
LHR-D576G	15.1 ± 0.9 (5.8-fold)	94.8 (72.9 to 135.5)	40.6 ± 2.1 (0.62-fold)
LHR-R476H	1.7 ± 0.5	- *^d^*	- *^d^*

Values are the means ± SD of a single representative experiment performed in triplicate. The half-maximal effective concentration (EC_50_) values were determined from the concentration-response curves from in vitro bioassays. *^a^* Basal cAMP level average without agonist treatment. *^b^* Rmax average cAMP level/10^4^ cells. *^c^* Geometric mean (95% confidence limit). *^d^* Nondetectable.

**Table 2 ijms-24-09133-t002:** Rates of cell-surface loss of receptors in transiently transfected cell lines expressing the wild-type eel LHR and mutants thereof.

Ligand Treatment	Eel LHR Cell Lines	t_1/2_ (min)	Plateau (% of Control)
rec-eel LH	eel LHR-WT	0.9 ± 0.1	64.2 ± 1.5
	eel LHR-D576G	0.6 ± 0.1	59.8 ± 2.5
	eel LHR-R476H	0.6 ± 0.1	61.9 ± 2.1
rec-eCG	eel LHR-WT	2.5 ± 0.2	56.8 ± 1.4
	eel LHR-D576G	2.1 ± 0.1	46.3 ± 1.0
	eel LHR-R476H	1.9 ± 0.1	42.6 ± 1.2

Data were fitted to one-phase exponential decay curves to obtain values of t_1/2_ and plateau (i.e., maximum reduction). The data were from three individual experiments.

## Data Availability

Not applicable.

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
