# Peer review of "Specific Signal Transduction of Constitutively Activating (D576G) and Inactivating (R476H) Mutants of Agonist-Stimulated Luteinizing Hormone Receptor in Eel"

_ijms, 2023, doi:10.3390/ijms24119133_

Round 1

Reviewer 1 Report (New Reviewer)

In this paper, authors established activating (D576G) and inactivating (R476) mutants of eel LHR to detect their effects on receptor signaling mechanisms. Which provided important information for the structure–function relationship of LHR–LH complexes.

1.     In figure 1, can author point out in Western blot figure which bands represent LHRs, moreover I hardly can see any signal in R476H channel.

2.     In figure 4, the data of WT group had been repeatedly used in 2 figures. I think authors integrate 2 figures into one.

3.     In figure 7, the data of WT group had been repeatedly used in 2 figures. I think authors integrate 2 figures into one.

4.     In this paper, I found a lot of hyphen had been used in wrong places. Authors should really double check and review their manuscript before submission.

Author Response

Reviewer 1

In this paper, authors established activating (D576G) and inactivating (R476) mutants of eel LHR to detect their effects on receptor signaling mechanisms. Which provided important information for the structure–function relationship of LHR–LH complexes.

  1. In figure 1, can author point out in Western blot figure which bands represent LHRs, moreover I hardly can see any signal in R476H channel.

®We carried out western blot analysis in the cell lysis expressing eel LHR-wt and mutants. The experiments were conducted several times, but the amount of expression in R476H mutant is very small on the cell surface. Therefore, we cannot detect any specific band of 70, 90 and 180 kDa in the Figure 2.

  1. In figure 4, the data of WT group had been repeatedly used in 2 figures. I think authors integrate 2 figures into one.

®We also thought that 2 figures combine 1 figure. In this figure, we want to clearly present the different of the cAMP responsiveness. Thus, we separate out 2 figures. In the eelLHR-D578G mutant, we also highlight the basal cAMP response without agonist treatment. R476H mutant did not any cAMP response.

Therefore, we suggest 2 separate figures in the figure 4.

  1. In figure 7, the data of WT group had been repeatedly used in 2 figures. I think authors integrate 2 figures into one.

®We first thought that 2 figures combine 1 figure as reviewer’s comment. However, the difference both D576G and R476H don’t identify much more loss in the surface receptor. If the figures come together 1 figure, the loss of surface receptor hard to draw a line in both mutants. Thus, we separate figures in the figure 7 as the figure 4..

  1. In this paper, I found a lot of hyphen had been used in wrong places. Authors should really double check and review their manuscript before submission.

®We checked all sentence by English Editing company by reviewer’s comment.

We also rechecked hyphens used in the manuscript. We tried to minimize the hyphens.

Reviewer 2 Report (New Reviewer)

The manuscript by Choi et al generated activating (D576G) and inactivating (R476) mutants of eel LHR to elucidate their effects on receptor signaling mechanisms, and cell surface loss of their receptors. However, apart form changes in the model system, this paper lacks novelty. Also, I have few question which author should address before considering this mansucript for publication.

1. How the cell surface expression of eel LHR was altered by single point mutation. What would be the cell surface expression of eel LHR upon double mutation (D576G and R476H).

2. Authors have mentioned in the manuscript stating that there are no suitable antibodies available for VSVG tag for flow cytometry, did authors tried FITC Anti-VSV-G tag antibody (ab3863) from abcam (https://www.abcam.com/products/primary-antibodies/fitc-vsv-g-tag-antibody-ab3863.html)?

3. What would be cAMP levels stimulated by recombinant eel-LH upon double D576G and R476H.

Author Response

Reviewer 2

The manuscript by Choi et al generated activating (D576G) and inactivating (R476) mutants of eel LHR to elucidate their effects on receptor signaling mechanisms, and cell surface loss of their receptors. However, apart form changes in the model system, this paper lacks novelty. Also, I have few question which author should address before considering this manuscript for publication.

  1. How the cell surface expression of eel LHR was altered by single point mutation. What would be the cell surface expression of eel LHR upon double mutation (D576G and R476H).

®The double mutants of activating and inactivating LHR may be low expression of cell surface for conformational change. Thus, R476H mutant is presumed to be low in the expression of cell surface receptors despite a little higher expression in the D576G mutant. Thus, we inserted in the discussion part “Although we did not check in the double mutants, the mutants may be low expression of cell surface for conformational change.” In the discussion.

  1. Authors have mentioned in the manuscript stating that there are no suitable antibodies available for VSVG tag for flow cytometry, did authors tried FITC Anti-VSV-G tag antibody (ab3863) from abcam (https://www.abcam.com/products/primary-antibodies/fitc-vsv-g-tag-antibody-ab3863.html)?

®We checked the anti-VSV-G tag antibodies .in the abcam and the other company. In the first reviewer’s comment, the reviewer suggested that the monoclonal antibody of VSVG-tag should be used in the flow cytometry experiment. Thus, we could not find the correct antibody. We also thought the ab50559 of mouse monoclonal antibody to VSVG tag. But this antibody is adequate to WB, IP, ICC.

®We deleted “However, we were unable to process the flow cytometry analysis on the receptor expression because there is no suitable antibody of VSVG-tag.’ In the result.

  1. What would be cAMP levels stimulated by recombinant eel-LH upon double D576G and R476H.

®Although we did not check cAMP response in the double mutants, it supposes to be a completely flat for the low expression of the cell surface receptor.

-We inserted “Even though we didn’t analyze it, the cAMP responsiveness will be at a completely low level in the double mutants of activating and inactivating eel LHR.” In the discussion.

Round 2

Reviewer 2 Report (New Reviewer)

No adequate justification for my concerns.  

Antibody ab3863 from Abcam is suitable for the flow cytometry experiment, I assume the authors did miss my comment and responded to comments with the wrong catalogue number (ab50559). The ab50559 antibody is not anti-VSV-G.  

Author Response

Reviewer 2

Antibody ab3863 from Abcam is suitable for the flow cytometry experiment, I assume the authors did miss my comment and responded to comments with the wrong catalogue number (ab50559). The ab50559 antibody is not anti-VSV-G.  

®Yes, I found that the antibody number was wrong in our previous revised content. The number (ab50549; mouse monoclonal antibody of VSVg tag) instead of 50559 should be presented. Thus, we realized that the previous revised content was not available. At that time, we didn’t find the antibody (ab3836) suggested in the previous comment. And we don’t have enough time to revise our paper within 10 days. After the first review, we determined that the selected antibody (a rabbit anti-VSVg tag monoclonal antibody; Cell signaling) is to do western blotting of expressing receptors instead of flow cytometry experiment. Thus, we added the western blotting experiment despite not enough to a satisfactory extent. The result makes a difference in the expression level of the cell surface. Therefore, we determined that results of between western blotting and cell surface expression of receptors were very similar.

This manuscript is a resubmission of an earlier submission. The following is a list of the peer review reports and author responses from that submission.

Round 1

Reviewer 1 Report

The topic is certainly interesting and obtained results may expand the knowledge of the scientific community regarding the topic, however the study has numerous shortcomings that make it inadequate for publication in the journal without a thorough revision.

In particular, the rationale for the study (line 68-71) should be more explicitly expressed and the language must be revised by a native English speaker. Moreover, while the part concerning the cAMP responsiveness is clear, that concerning the loss of receptor’s expression after agonist treatment must be clearer in both  methods and results section.

In my opinion, the expression of mutated receptor on the cell surface must be determined by a more appropriate method such as fluorescence activated cell sorting (FACS analysis), using a monoclonal antibody also after cell permeabilization to detect the amount of the receptor trapped inside the cells. Binding studies to determine the total number of receptors expressed at the surface of cell transfected with the mutants and their relative dissociation constant, may also be useful. For this, further experiments are recommended.

The number of  self-citations must be reduced.

The description of Figure 5 is not present in the text.

Possible typing errors must be corrected all over the text.

Author Response

Reviewer 1

The topic is certainly interesting and obtained results may expand the knowledge of the scientific community regarding the topic, however the study has numerous shortcomings that make it inadequate for publication in the journal without a thorough revision.

In particular, the rationale for the study (line 68-71) should be more explicitly expressed and the language must be revised by a native English speaker.

 ®We changed “Line 68-71” to “Although several studies have been reported on signal transduction of naturally occurring hLHR, rLHR, and eLH/CGR, there are few reports on the characterization and function of signal transduction of eel LHR in activating/inactivating mutants. We designed the present study to investigate the possibility that the highly conserved activating/inactivating mutants in eel LHR are implicated in signal transduction and loss of cell-surface receptors in cells expressing these receptors.

-Moreover, while the part concerning the cAMP responsiveness is clear, that concerning the loss of receptor’s expression after agonist treatment must be clearer in both methods and results section.

 ®We inserted “As a result, we observed a specific increase in basal cAMP response by the cells expressing eel LHR-D576G mutant. However, the inactivating mutant (eel LHR-R476H) completely impaired the cAMP response.” In the cAMP section on page 5.

In my opinion, the expression of mutated receptor on the cell surface must be determined by a more appropriate method such as fluorescence activated cell sorting (FACS analysis), using a monoclonal antibody also after cell permeabilization to detect the amount of the receptor trapped inside the cells. Binding studies to determine the total number of receptors expressed at the surface of cell transfected with the mutants and their relative dissociation constant, may also be useful. For this, further experiments are recommended.

 ®We conducted the expression levels many times in this study. Although we did not do binding assay and FACS analysis, we confirmed that the result of the expression levels in the presented study is almost accurate for many papers published in our previously results.

The number of self-citations must be reduced.

 ®We reduced the number of self-citations and reference are reduced.

The description of Figure 5 is not present in the text.

 ®We inserted “Figure 5” in the text page 4 and page 5.

Possible typing errors must be corrected all over the text.

 ®We corrected all sentences to the reviewer’s comments and re-edited by the English editing company

Reviewer 2 Report

Choi et al. Explored Specific Signal Transduction of Constitutively Activating (D576G) 2 and Inactivating (R476H) Mutants of Agonist-stimulated Luteinizing Hormone Receptor in Eel

Introduction section is well written

Results section:

I see no green circles in figure 1, please revise

Data should be presented as mean and SD, not seem to avoid confusion. A lot of authors neglects the difference between SEM and SD, and uses SEM as results seem more precise.

Please specify the statistical test used in each of the figures, as these represent independent parts of the manuscript.

Figure 6 and 7 should be revised as these are hardly readable (perhaps make them larger)

P values indicating differences between different bars should be included in each bar chart

Discussion

Discussion section should be revised, specifically with regards to applicability of such results

In addition, the limitations of study should be addressed

Methods

I wonder if the authors checked the normality of data distribution.

Overall, the study is well written, but it would benefit from more skilfully conducted discussion and addressing some methodological issues.

Author Response

Reviewer 2

Introduction section is well written

Results section:

I see no green circles in figure 1, please revise

 ®We changed “green” to “blue” in the Figure 1 legend.

Data should be presented as mean and SD, not seem to avoid confusion. A lot of authors neglects the difference between SEM and SD, and uses SEM as results seem more precise.

 ®We changed “SEM” to “S.D” as reviewer’s comments.

Please specify the statistical test used in each of the figures, as these represent independent parts of the manuscript.

 ®We changed and inserted in all figures by reviewer’s comments.

Figure 6 and 7 should be revised as these are hardly readable (perhaps make them larger)

 ®We changed more larger

P values indicating differences between different bars should be included in each bar chart

®We inserted P value difference between groups in each Bar graph.

Discussion

Discussion section should be revised, specifically with regards to applicability of such results

 ®We inserted “Mutations in LHRs and FSHRs could affect receptor trafficking to the plasma membrane and cause disorders in fertility. Thus, the role of LHRs in male gonad development is crucial. A significant challenge from a pharmacologic point of view is how to diminish cAMP signaling by constitutively activating LHR without agonist treatment” in the Discussion section.

In addition, the limitations of study should be addressed

 ®We inserted “In the presented study, we did not determine the internalization in cells expressing both active and inactive receptors. However, we suggest that the receptor loss from the cell surface could help predict the internalization rate into the endosome.” In the Discussion section

®We inserted “Thus, we must accurately resolve these unexplained problems activating or inactivating eel LH receptors. Research is currently underway to prove our theory about the internalization, degradation, and recycling of GPCR.” In the Discussion section

Methods

I wonder if the authors checked the normality of data distribution.

Overall, the study is well written, but it would benefit from more skillfully conducted discussion and addressing some methodological issues.

  ®We rerranged the data of the experimental procedure.

Round 2

Reviewer 1 Report

Despite the authors' appreciable effort to improve the quality of the manuscript in object, in my opinion to perform further experiments concerning the expression of mutated receptors on the cell surface are fundamental to make manuscript publishable in the journal.

I therefore suggest to perform the required experiments by more appropriate methods and re-submit the manuscript.